# Psychological distress during the COVID-19 epidemic in Chile: The role of economic uncertainty

Fabián Duarte[1,2]*, Álvaro Jiménez-Molina[2,3,4,5]

1 Department of Economics, Facultad de Economía y Negocios, Universidad de Chile, Santiago, Chile,
2 Millennium Nucleus in Social Development (DESOC), Santiago, Chile, 3 Millennium Nucleus to Improve the Mental Health of Adolescents and Youths (Imhay), Santiago, Chile, 4 Millennium Institute for Research in Depression and Personality (MIDAP), Santiago, Chile, 5 Facultad de Psicología, Universidad Diego Portales, Santiago, Chile

* fabduarte@fen.uchile.cl

**Data Availability Statement:** All relevant data are within the manuscript and its Supporting Information files.

**Funding:** FD and AJ-M received funding from ANID/Millennium Science Initiative, grant

## Abstract

Previous research has shown that the COVID-19 outbreak, social distancing, and lockdown can affect people's psychological well-being. The aims of this study were (1) to estimate the extent to which perceptions and expectations regarding the social, economic, and domestic effects of the COVID-19 outbreak are associated with psychological distress and (2) to identify some demographic, psychosocial, and economic factors associated with increased vulnerability to psychological distress during the COVID-19 outbreak in Chile. 1078 people participated in a telephone survey between May 30 and June 10, 2020. The sample is representative of the Chilean adult population. Psychological distress was assessed through a questionnaire of anxious and depressive symptoms (Patient Health Questionnaire-4). We analyzed the data set using ordinary least-squares regression models, first estimating models for the entire sample, and then stratifying the sample into different groups to explore differences by gender and age. 19.2% of participants displayed significant psychological distress (PHQ-4 $\geq$ 6), with moderate to severe anxiety-depression symptoms being more prevalent in women than in men (23.9% vs 14.1%, $\chi^2$ 16.78, p<0.001). The results of this study suggest that being a woman, feeling lonely and isolated, living in the areas hit hardest by the pandemic and lockdown, expecting a lack of income due to having to stop working as a consequence of the pandemic, and having a history of diagnosed mental disorders are significantly associated with psychological distress (p<0.05). The results of this study highlight the need to implement psychosocial programs to guard people's psychological well-being and social policies to address economic uncertainty during the current COVID-19 outbreak in Chile.

## Introduction

The COVID-19 pandemic has produced a sudden change in people's life, which could result in tensions at the personal, family, and community levels. Over the past few months, several studies have suggested that the disruption to social life derived from the current pandemic and its

"Millennium Nucleus in Social Development (DESOC), NCS17_015". AJ-M also received funding from ANID/Millennium Science Initiative, grant "Millennium Nucleus to Improve the Mental Health of Adolescents and Youths (Imhay), NCS17_035", and "Millennium Institute for Research in Depression and Personality (MIDAP), ICS13_005", and ANID/FONDECYT POSTDOCTORADO/2020-3200944. The funders had no role in study design, data collection and analysis, decision to publish, or preparation of the manuscript.

**Competing interests:** The authors have declared that no competing interests exist.

mitigation measures, along with increased economic and psychosocial stress, may significantly affect the quality of life and mental health of the population in the short, medium, and long term [1–5].

Social distancing and lockdown measures to contain the spread of COVID-19 have radically reduced several sources of social contact, depriving many people of the social support they can normally rely on to deal with economic and psychosocial stressors [2]. Although the results of emergent studies are heterogeneous, it has been suggested that social distancing and lockdown may be associated with a sense of isolation, in addition to producing stress, anxiety, sadness, irritability, insomnia, and feelings of helplessness [1, 2, 4–6]. While these reactions are expected and even adaptive to situations involving disruptions to normal life and close encounters with illness and death, when they become intense or prolonged over time, they can have mental health consequences such as depression, anxiety, and increased suicide rates [2, 4–8]. Likewise, some studies indicate that while social distancing and lockdown affect the entire population, these measures can be even more stressful for women [4, 9, 10], young people [4, 11, 12], and people with pre-existing physical and mental health problems [1–5, 13].

The COVID-19 outbreak has created a widespread atmosphere of unpredictability and uncertainty about the future. Similarly, after the financial crisis of 2008, several studies have shown the adverse impact of economic uncertainty on psychological well-being [14–16]. Job insecurity and unemployment, fear of reduced income and increased debt, food insecurity, and the unavailability of basic supplies have a significant impact on psychological well-being [17–19], which can manifest itself as anxiety and depressive symptoms [20–22]. In other words, the economic breakdown resulting from the COVID-19 pandemic could increase the risk of psychological distress and exacerbate health inequalities [3, 22].

In the emergent literature on COVID-19, some studies have focused on people's perceptions and attitudes towards the pandemic and its negative consequences on the economy and health [22–26]. For example, Cerami et al. [25] evaluated the perceived impact of the COVID-19 outbreak on the health and economy of the Italian population during the early days of extreme social distancing measures. They suggest that individual differences in the perceived health impact of the COVID-19 outbreak are modulated by psychosocial fragility, especially distress and loneliness [25]. Likewise, Codagnone et al. [26] conducted a multi-site study to estimate the consequences of the economic crisis on mental health in Italy, Spain, and the United Kingdom. Their results suggest that around 43% of the population in these three countries is at high risk of suffering from stress, anxiety, and depressive symptoms associated with being economically vulnerable and having been affected by a negative economic shock.

Globally, most studies on COVID-19 and psychological well-being have been conducted in high-income countries [4]. However, it has been suggested that the economic and mental health effects of COVID-19 could be worse in developing regions [6, 27] and in states with weak social safety nets [22, 27], such as Latin America. Despite the implementation of strict mitigation measures, the pre-pandemic conditions that characterize Latin American countries (high levels of poverty, inequality, informal employment, and vulnerable populations with limited access to social welfare) have undermined the effectiveness of their responses to the pandemic [28].

Chile is a mid-to-high-income country characterized by persistent and comparatively high income inequality [29]. Since mid-March, the country has rapidly moved towards severe restrictions on movement and social contact due to a combined scenario of high COVID-19 incidence and mortality rates [28]. Since then, a large percentage of the population has faced serious economic difficulties [28, 30]. An early exploratory study on the perceived impact and future concerns regarding the COVID-19 lockdown in Santiago showed that a large number of people (49.8%) were concerned about economic issues (e.g. income reduction) during the

first two weeks of quarantine [31]. In this context, the Chilean government launched a food supply campaign at the end of May and announced an "emergency family cash transfer" (approximately US$120 per person over a four-month period) for the benefit of the most affected households [28, 32]. In addition, at the end of July, the National Congress passed a law allowing workers to withdraw up to 10% of their pension funds to respond to the crisis. All these measures are aimed at alleviating growing unemployment and the impact of the COVID-19 outbreak on the household economy.

In this context, the aim of this study was to estimate the extent to which perceptions and expectations regarding the social, economic, and domestic effects of the COVID-19 outbreak are associated with psychological distress in the Chilean population. Using data from a representative telephone survey of the adult population in the country, we identify demographic, psychosocial, and economic factors associated with increased vulnerability to psychological distress during the COVID-19 outbreak in Chile.

The relevance of this study is twofold. First, Chile is a country with a relatively high burden of mental illness and low access to mental health services [33]. Second, unlike most international studies on COVID-19 and psychological distress that have used non-probability sampling techniques [34], we offer an assessment based on a nationwide representative survey.

## Materials and methods

### Procedure

The data were collected through a telephone survey from May 30 to June 10, 2020. This period coincided with one of the most critical moments of the COVID-19 outbreak in Chile and the implementation of the most severe measures of social distancing and confinement. The sample is representative of the adult population in Chile. The instrument used comprises a variety of questions related to psychological distress, perceptions of the social and family impact of COVID-19, economic expectations, and sociodemographic variables.

The construction of the sample was multi-stage. The first stage emerged from a probabilistic and geographically stratified sampling strategy that randomly selects municipalities, then census blocks and occupied dwellings, and finally people aged 18 years or more. In this process, the expansion factor was calculated as the inverse of the probability of selection or inclusion in the sample. It also includes adjustments for telephone non-response and post-stratification adjustments for age ranges (18–35, 36–59, ≥60) of men and women, in two stages. Both corrections are made to reduce telephone survey bias in accordance with ECLAC recommendations [35].

The contact rate was 39.9% with a cooperation rate of 60.7%, so the response rate was 24.2% of the initial sample. With these values and assuming simple random sampling, and for a proportion of 0.5, the survey is nationally representative (with an absolute error of 2.98%) for men and women (with absolute errors of 4.31% and 4.14%, respectively) and for age groups 18–35, 36–59, and ≥60 (with absolute errors of 5.77%, 4.49%, and 5.57%, respectively).

### Participants

The survey was answered by 1,078 people living in urban and rural areas in all 16 regions of the country. With respect to the sociodemographic characteristics of the sample (Table 1), 48.9% of the participants were men and 51.1% were women, with an average age of 44 years (SD 17). Most of the participants (71.3%) lived in central Chile, where the Metropolitan Region (57% of participants) and the capital, Santiago, are located. The average educational level was 13 years of formal education (SD 3.56) and the average income was USD$ 922.

**Table 1. Sociodemographic characteristics of the sample.**

| Variables | n (%) | weighted population |
|---|---|---|
| **Sex** | | |
| Male | 527 (48.9) | 7320604 |
| Female | 551 (51.1) | 7661738 |
| **Age** | | |
| 18–35 | 396 (36.7) | 5502491 |
| 36–59 | 441 (40.9) | 6131842 |
| ≥60 | 241 (22.4) | 3348010 |
| **Household Income (USD)** | 865* (SD 883) | |
| **Education** | | |
| Primary school (≤8 years) | 121 (11.2) | 1687482 |
| High school (between 9 and ≤12 years) | 440 (40.9) | 6126840 |
| Higher education (>13 years) | 515 (47.8) | 7168021 |
| **Geographic Area (place of residence)** | | |
| Northern Chile | 83 (7.7) | 1150655 |
| Central Chile | 768 (71.2) | 10675554 |
| Southern Chile | 227 (21.0) | 3156134 |

* mean.

US$1 = CLP$807 (average May and June 2020).

## Ethics approval and consent

Ethical approval was obtained from the Ethics Committee of the Faculty of Economics and Business of the University of Chile. Because the survey was conducted by telephone, the consent form was read by the interviewers and participants gave their verbal consent before responding, which was audio recorded. The ethics committee approved this verbal consent procedure.

## Measures

**Psychological distress.** We operationalized psychological distress through the Patient Health Questionnaire-4 (PHQ-4) [36], an ultra-brief screening tool (4-item self-report questionnaire) used for the evaluation of depressive and generalized anxiety symptoms according to criteria set out in the fourth edition of the Diagnostic and Statistical Manual of Mental Disorders (DSM-IV). The questions on depressive symptoms (PHQ-2) [37] point to the two cardinal symptoms of depression (depressed mood and anhedonia) defined in the Patient Health Questionnaire-9 (PHQ-9) [38], while the two questions on anxiety point to the cardinal symptoms of generalized anxiety disorder defined in the Generalized Anxiety Disorder-7 questionnaire (GAD-7) [39].

The PHQ-4 is composed of a 4-point ordinal response scale (from 0 = not at all to 3 = nearly every day). A total score ≥ 6 indicates the presence of moderate to severe anxiety-depression symptoms. Although the PHQ-4 is not a diagnostic instrument, it has demonstrated good accuracy for detecting both anxiety and depressive disorders [36]. Likewise, a recent meta-analysis shows that the sensitivity of the PHQ-2 is higher than that of semi-structured interviews, which are close to clinical interviews conducted by professionals trained in the diagnosis of major depression [40]. In this sample, the PHQ-4 had an internal consistency of 0.78.

We evaluated psychological distress through anxiety and depression symptoms because they are among the most common mental health problems and often co-occur and overlap

[41]. Since psychological distress exists along a continuum ranging from mild, time-limited distress to severe mental health conditions, the dependent variables were regarded as continuous.

**Physical health status.**   We assessed the presence of a physical health condition through the following question: "Are you currently diagnosed with any of the following diseases or health conditions? [Hypertension, Obesity, Diabetes]" [0 = No, 1 = Yes].

**Mental health diagnosis and treatment.**   We also evaluated a history of mental disorder at some point in the lifespan ("At some point in your life, have you been diagnosed with an illness such as depression, anxiety disorder, bipolar disorder, or another mental health problem?" [0 = No, 1 = Yes]) and mental health treatment during the past 12 months ("During the last 12 months, have you been in treatment for any mental health problems?" [0 = No, 1 = Yes]).

**Loneliness.**   To measure loneliness, we used an adapted version of the Three-Item Loneliness Scale [42], a short scale developed specifically to assess perceptions of social isolation through a telephone survey. The three items on the scale were merged into one question: "Over the last two weeks, how often did you feel that you lacked companionship, that you were being left out, or that you were isolated from others?" (0 = not at all to 3 = nearly every day).

In addition, several variables of perception and expectation were evaluated through questions about the economy and the future of the household, given the pandemic and the lockdown measures in place.

**Space at home, basic supplies, domestic conflict, income expectation.**   "If the area where your home is located is placed on lockdown, which of the following difficulties do you think you or other members of your household will have to face?: A lack of space at home, Difficulty in accessing food and basic supplies, Conflict within the household, A lack of income due to having to stop working" [0 = No, 1 = Yes]).

**Household debt expectation.**   "Do you think your household's debt situation over the next three months will be. . .? [1 = Better, 2 = Same, 3 = Worse].

Finally, demographic and socioeconomic variables were included. These variables were used in the model as continuous variables (age and log of income) and dummy variables (gender, geographic area, education).

## Data analysis

Gender differences in psychosocial and health variables were calculated using the $\chi^2$ test. To better understand the association between psychological distress and the control variables, we used an ordinary least-squares regression. Expansion factors were included in the calculations of the regression models. In regression models, we considered depressive symptoms (from 0 to 6 points), anxiety symptoms (from 0 to 6 points), and psychological distress (depressive symptoms + anxiety symptoms, from 0 to 12 points) as separate dependent variables. To observe the possible differences by sex and age category, we controlled for these variables, as well as for the log of income, geographical location, education level, and the variables defined in the previous section (Loneliness, Physical health status, Mental health diagnosis and treatment, Space at home, Basic supplies, Domestic conflict, Income expectation, and Household debt expectation).

## Results

As Table 2 indicates, we found that 207 (19.2%) participants displayed significant psychological distress (PHQ-4 $\geq$ 6). The results show a statistically significant difference by sex, with moderate to severe anxiety-depression symptoms being more prevalent in women than in men (23.9% vs 14.09%, $p < 0.001$). Likewise, 21.15% of the participants mentioned having been diagnosed with a mental disorder at some point in their lives, while 9.94% reported

**Table 2. Psychosocial and health variables (n = 1078).**

| Variables | Total (%) | Female (%) | Male (%) | $\chi^2$ test | p value |
|---|---|---|---|---|---|
| Psychological distress | 19.20 | 23.90 | 14.09 | 16.78 | 0.000 |
| Mental health diagnosis | 21.15 | 28.39 | 13.32 | 36.66 | 0.000 |
| Mental health treatment | 9.94 | 13.98 | 5.60 | 21.07 | 0.000 |
| Physical health status (hypertension, obesity and/or diabetes) | 38.31 | 43.39 | 32.82 | 12.73 | 0.000 |
| Loneliness | 11.81 | 14.49 | 8.91 | 8.01 | 0.005 |
| Expectation of lack of income | 47.59 | 46.25 | 49.03 | 0.84 | 0.360 |
| Expectation of increased debt | 47.31 | 47.50 | 47.10 | 0.017 | 0.897 |
| Expectation of difficulty getting food | 36.18 | 36.25 | 36.10 | 0.00 | 0.959 |
| Expectation of increased conflict in home | 8.44 | 6.79 | 10.23 | 4.13 | 0.042 |
| Lack of space in the home | 13.82 | 12.32 | 15.44 | 2.20 | 0.138 |

having accessed mental health treatment during the last 12 months, again with a higher proportion of women in both dimensions (p < 0.001).

It was also observed that 38.31% of the participants suffered from hypertension, obesity, and/or diabetes. Moreover, 11.81% of participants had felt excluded or isolated from others during the last two weeks.

In relation to economic and domestic expectations in the context of the COVID-19 outbreak, 47.59% of participants think they will face a lack of income due to having to stop working, 36.18% anticipate difficulties in obtaining food and basic supplies, and 48.7% think that household debt will increase over the next three months. In addition, 8.44% of participants think that conflict in their home will increase, while 13.82% mention that they will face a lack of space in the home. As shown in Table 2, most of the differences between women and men are significant (p < 0.01).

The results of the linear regression models showed that being a woman, feeling lonely or isolated, living in a specific geographic area (central Chile), and expecting a lack of income due to having to stop working were significantly associated (p < 0.05) with psychological distress (Table 3).

Regarding specific models by symptom category, being a woman, feeling lonely or isolated, living in central Chile, and having a mental health diagnosis were significantly associated with anxiety symptoms, while female gender, age, geographic area of residence (central Chile), feelings of loneliness or isolation, and expectations of a lack of income due to having to stop working were significantly associated with depressive symptoms.

Since regression models revealed differences by sex and age, we generated alternative models stratified by sex and age category. The alternative models stratified by sex (Table 4) showed that, in the case of women, feelings of loneliness or isolation and a prior diagnosis of mental disorder were the only variables significantly associated with psychological distress, while in the case of men, in addition to feelings of loneliness or isolation, living in central Chile was associated with psychological distress.

It is interesting to note that, in these sex-stratified models, household income was significantly associated with depressive symptoms. However, this association was positive for women and negative for men. In addition, in the case of men, there was a negative association between age and depressive symptoms.

We also present alternative regression models differentiated by age category (Table 5). Overall, the results of these models show that, in the case of young people (aged 18–35), only feelings of loneliness and isolation were significantly associated with psychological distress. In the case of people aged 36–59, it is interesting to note that higher education was negatively

**Table 3. Regression models.**

| Variables | Anxiety symptoms | Depressive symptoms | Psychological distress |
|---|---|---|---|
| Loneliness | 0.376*** | 0.661*** | 1.037*** |
| | (0.164–0.588) | (0.473–0.850) | (0.737–1.337) |
| Female | 0.527*** | 0.499*** | 1.024*** |
| | (0.207–0.847) | (0.136–0.862) | (0.441–1.608) |
| Center | 0.368** | 0.288** | 0.656*** |
| | (0.0647–0.672) | (0.00401–0.572) | (0.159–1.154) |
| South | 0.191 | 0.261** | 0.453* |
| | (-0.112–0.494) | (0.00212–0.521) | (-0.0346–0.942) |
| Age 36–59 | 0.219 | -0.469** | -0.251 |
| | (-0.133–0.572) | (-0.850–-0.0889) | (-0.881–0.380) |
| Age 60+ | 0.531** | -0.545** | -0.0143 |
| | (0.0541–1.007) | (-1.089–-0.00142) | (-0.884–0.855) |
| Ln household income | -0.0264 | -0.0105 | -0.0364 |
| | (-0.282–0.229) | (-0.273–0.252) | (-0.500–0.427) |
| Expectation of lack of income | 0.346* | 0.478*** | 0.824*** |
| | (-0.00259–0.695) | (0.123–0.833) | (0.226–1.422) |
| Expectation of difficulty getting food | -0.0588 | 0.0137 | -0.0438 |
| | (-0.418–0.301) | (-0.371–0.398) | (-0.686–0.598) |
| Increased expectation of conflict in home | 0.475* | 0.220 | 0.695 |
| | (-0.0889–1.039) | (-0.318–0.758) | (-0.317–1.706) |
| Lack of space in the home | 0.189 | 0.190 | 0.378 |
| | (-0.330–0.708) | (-0.264–0.645) | (-0.507–1.263) |
| Expectation of increased debt | 0.134 | 0.124 | 0.256 |
| | (-0.249–0.516) | (-0.231–0.479) | (-0.381–0.894) |
| Physical health status (hypertension, obesity and/or diabetes) | -0.149 | -0.169 | -0.316 |
| | (-0.456–0.157) | (-0.463–0.125) | (-0.813–0.180) |
| Mental health diagnosis | 0.540** | 0.110 | 0.652* |
| | (0.101–0.978) | (-0.274–0.493) | (-0.0716–1.375) |
| Mental health treatment | 0.376 | 0.0451 | 0.419 |
| | (-0.208–0.960) | (-0.505–0.596) | (-0.561–1.399) |
| High school or lower | -0.157 | 0.102 | -0.0575 |
| | (-0.670–0.357) | (-0.251–0.455) | (-0.779–0.664) |
| Higher education | -0.234 | 0.180 | -0.0552 |
| | (-0.810–0.343) | (-0.295–0.655) | (-0.952–0.841) |
| Constant | 1.091 | 0.687 | 1.773 |
| | (-2.409–4.590) | (-2.959–4.334) | (-4.593–8.139) |
| Observations | 982 | 982 | 980 |
| R-squared | 0.164 | 0.243 | 0.219 |

Robust CI in parentheses.

*** p<0.01,

** p<0.05,

* p<0.1.

associated with psychological distress, while in the case of older people (age $\geq$ 60), the expectation of a lack of income was the variable with the strongest association with psychological distress.

**Table 4. Regression models by sex.**

| Variables | Male | | | Female | | |
|---|---|---|---|---|---|---|
| | Anxiety symptoms | Depressive symptoms | Psychological distress | Anxiety symptoms | Depressive symptoms | Psychological distress |
| Loneliness | 0.421** | 0.659*** | 1.080*** | 0.314*** | 0.633*** | 0.948*** |
| | (0.0426–0.798) | (0.418–0.901) | (0.606–1.555) | (0.107–0.521) | (0.408–0.858) | (0.587–1.309) |
| Center | 0.472** | 0.341* | 0.812** | 0.198 | 0.183 | 0.380 |
| | (0.0701–0.874) | (-0.0262–0.708) | (0.150–1.473) | (-0.240–0.636) | (-0.272–0.638) | (-0.356–1.115) |
| South | 0.459** | 0.236 | 0.696* | -0.115 | 0.162 | 0.0465 |
| | (0.0398–0.878) | (-0.169–0.642) | (-0.0282–1.420) | (-0.574–0.344) | (-0.201–0.526) | (-0.645–0.738) |
| Age | 0.00395 | -0.0274*** | -0.0235* | 0.0138** | -0.00981 | 0.00401 |
| | (-0.0117–0.0196) | (-0.0413–-0.0136) | (-0.0478–0.000841) | (0.00132–0.0263) | (-0.0270–0.00742) | (-0.0221–0.0301) |
| Ln income | -0.0970 | -0.409** | -0.507 | 0.0788 | 0.379** | 0.458* |
| | (-0.488–0.294) | (-0.739–-0.0793) | (-1.168–0.153) | (-0.199–0.357) | (0.0479–0.710) | (-0.0721–0.988) |
| Income reduction expectation | 0.271 | 0.302 | 0.574 | 0.322 | 0.396 | 0.721* |
| | (-0.253–0.794) | (-0.0872–0.692) | (-0.153–1.300) | (-0.107–0.752) | (-0.104–0.896) | (-0.106–1.548) |
| Expectation difficulty getting food | -0.137 | -0.397* | -0.537 | 0.0278 | 0.367 | 0.396 |
| | (-0.684–0.410) | (-0.859–0.0651) | (-1.448–0.374) | (-0.396–0.452) | (-0.138–0.872) | (-0.397–1.189) |
| Increased expectation of conflict in the home | 0.703* | 0.492 | 1.197* | 0.211 | -0.241 | -0.0303 |
| | (-0.0769–1.483) | (-0.190–1.174) | (-0.154–2.548) | (-0.428–0.851) | (-1.032–0.550) | (-1.281–1.220) |
| Lack of space in the home | 0.300 | 0.292 | 0.593 | -0.0608 | 0.0677 | 0.00700 |
| | (-0.466–1.067) | (-0.372–0.956) | (-0.705–1.891) | (-0.731–0.610) | (-0.408–0.543) | (-1.009–1.023) |
| Expectation of increasing debt | 0.195 | 0.0444 | 0.239 | 0.103 | 0.344 | 0.444 |
| | (-0.405–0.795) | (-0.406–0.494) | (-0.689–1.167) | (-0.342–0.547) | (-0.107–0.794) | (-0.344–1.232) |
| Physical health status | -0.0690 | -0.0704 | -0.141 | -0.222 | -0.285 | -0.505 |
| | (-0.509–0.370) | (-0.476–0.336) | (-0.851–0.569) | (-0.620–0.176) | (-0.654–0.0844) | (-1.136–0.125) |
| Mental health diagnosis | -0.225 | -0.438* | -0.667 | 1.159*** | 0.618** | 1.778*** |
| | (-0.808–0.358) | (-0.878–0.00230) | (-1.565–0.232) | (0.638–1.681) | (0.133–1.104) | (0.916–2.641) |
| Mental health treatment | 0.824* | -0.152 | 0.676 | 0.0491 | 0.0297 | 0.0787 |
| | (-0.0243–1.671) | (-0.954–0.650) | (-0.643–1.995) | (-0.648–0.746) | (-0.556–0.615) | (-1.064–1.221) |
| High school or lower | 0.183 | 0.341 | 0.523 | -0.489 | -0.115 | -0.605 |
| | (-0.431–0.797) | (-0.148–0.829) | (-0.459–1.504) | (-1.260–0.282) | (-0.617–0.387) | (-1.639–0.429) |
| Higher education | 0.000603 | 0.348 | 0.347 | -0.554 | -0.182 | -0.735 |
| | (-0.684–0.686) | (-0.209–0.904) | (-0.774–1.468) | (-1.383–0.275) | (-0.884–0.519) | (-1.970–0.500) |
| Constant | 1.728 | 6.987*** | 8.733* | 0.293 | -3.631 | -3.337 |
| | (-3.909–7.365) | (2.436–11.54) | (-0.469–17.93) | (-3.447–4.033) | (-8.152–0.890) | (-10.52–3.850) |
| Observations | 471 | 472 | 470 | 511 | 510 | 510 |
| R-squared | 0.157 | 0.344 | 0.262 | 0.181 | 0.267 | 0.232 |

Robust CI in parentheses.

*** p<0.01,

** p<0.05,

* p<0.1.

## Discussion

To our knowledge, this is the first nationwide representative survey of psychological distress during the COVID-19 outbreak in Chile. We found that 19.2% of participants displayed psychological distress, with moderate to severe anxiety-depression symptoms being more

**Table 5. Regression models of psychological distress by age group.**

| Variables | 18 to 35 | 36 to 59 | ≥60 |
|---|---|---|---|
| Loneliness | 0.975*** | 1.012*** | 1.023*** |
| | (0.459–1.490) | (0.528–1.495) | (0.449–1.596) |
| Female | 0.403 | 1.648*** | 1.266*** |
| | (-0.713–1.520) | (0.991–2.305) | (0.384–2.149) |
| Center | 0.512 | 0.912** | 0.887 |
| | (-0.423–1.446) | (0.147–1.678) | (-0.181–1.955) |
| South | 0.349 | 0.970** | 0.467 |
| | (-0.664–1.362) | (0.146–1.795) | (-0.499–1.433) |
| Age | -0.0750* | -0.00768 | -0.0696* |
| | (-0.157–0.00681) | (-0.0595–0.0441) | (-0.140–0.000700) |
| Ln income | -0.549 | 0.497* | -0.151 |
| | (-1.453–0.354) | (-0.0844–1.078) | (-0.761–0.458) |
| Income reduction expectation | 0.397 | 0.440 | 1.441*** |
| | (-0.665–1.460) | (-0.340–1.220) | (0.443–2.439) |
| Expectation difficulty getting food | 0.608 | -0.00789 | -0.784 |
| | (-0.753–1.969) | (-0.684–0.668) | (-1.972–0.405) |
| Increased expectation of conflict in the home | 0.527 | 0.977 | -0.280 |
| | (-1.094–2.148) | (-0.472–2.426) | (-2.635–2.074) |
| Lack of space in the home | 0.332 | 0.204 | 0.749 |
| | (-1.247–1.910) | (-0.813–1.221) | (-0.803–2.301) |
| Expectation of increasing debt | 0.0284 | 0.370 | 0.277 |
| | (-1.305–1.362) | (-0.372–1.113) | (-0.713–1.267) |
| Physical health status | -0.0482 | -0.366 | -0.0143 |
| | (-1.286–1.189) | (-1.061–0.329) | (-0.891–0.862) |
| Mental health diagnosis | 0.301 | 1.329** | -0.0479 |
| | (-0.515–1.117) | (0.130–2.527) | (-0.991–0.896) |
| Mental health treatment | 1.101 | 0.279 | -0.552 |
| | (-0.353–2.555) | (-1.207–1.766) | (-1.976–0.871) |
| High school or lower | -0.432 | -0.460 | 0.160 |
| | (-3.215–2.352) | (-1.491–0.571) | (-0.974–1.294) |
| Higher education | 0.139 | -1.242** | 0.0646 |
| | (-2.834–3.113) | (-2.420–-0.0633) | (-1.267–1.396) |
| Constant | 11.16* | -5.047 | 7.572 |
| | (-1.387–23.70) | (-13.80–3.705) | (-2.459–17.60) |
| Observations | 262 | 436 | 282 |
| R-squared | 0.236 | 0.302 | 0.308 |

Robust CI in parentheses.

*** $p < 0.01$,

** $p < 0.05$,

* $p < 0.1$.

prevalent in women than men (23.9% vs 14.1%). Although we do not have a pre-pandemic baseline in the same sample, we can add that 43.9% of participants perceived their current mood to be "worse" or "much worse" compared with their mood prior to the social distancing and lockdown measures due to the COVID-19 outbreak, and that this perceived impact was significantly higher in women than in men (48.9% vs 38.4%, $\chi^2$ test 12.07, p = 0.001). In

addition, 42.4% of men stated that their mood had not changed due to the pandemic and confinement, in contrast to 28.7% of women (unreported result).

In this regard, it is interesting to add that, when comparing our results with a pre-pandemic longitudinal survey based on a Chilean representative community sample (ELSOC, https://coes.cl/encuesta-panel/), we observed a higher prevalence of depressive symptoms during 2020. The ELSOC study shows that, between 2016 and 2018, the average prevalence of depressive symptoms (PHQ-2 $\geq$ 3) was 20.6% (24.1% women vs 16.9% men). In our study, in contrast, this prevalence was 23.3% (30.8% women vs 15.5% men), which could suggest an increase in the burden of depressive symptoms in the population due to the pandemic. When comparing prevalence by age, the burden of depressive symptoms reported in the ELSOC for the 2016–2018 period was 21.2% (18–35 years), 20.1% (36–59 years), and 20.6% (60+ years). In our study, these prevalences were 31.5%, 18.5%, and 18.5%, respectively. These results suggest that, during the pandemic, there may have been an increased burden of depressive symptoms among women and young people. However, this comparison should be treated with caution, as the pre-pandemic estimates come from a different sample.

The results of this study show that being a woman, feeling lonely and isolated, living in the areas hit hardest by the pandemic and lockdown (central Chile), expecting a lack of income due to having to stop working, and having a prior mental health diagnosis are significantly associated with psychological distress.

Previous studies have suggested that social distancing and lockdown measures to contain the spread of the COVID-19 outbreak could be associated with a sense of loneliness and isolation, and that deprivation of social contact can translate into interpersonal stress and depressive symptoms, affecting psychological well-being [1–5, 43].

A relevant finding of this study is that almost half of the participants think they will face a lack of income due to having to stop working as a direct or indirect effect of the epidemic outbreak, and that this expectation has a significant effect on their psychological distress. Previous studies show that economic uncertainty is a common experience in the current pandemic context [22, 25, 44]. Likewise, a study conducted in Santiago during the first two weeks of the pandemic highlighted that economic uncertainty and fear of income reduction were among people's main concerns [31]. Pre-pandemic studies have suggested that coping with social crises can increase people's tendency to focus on the consequences for the household economy [45]. Feelings of economic uncertainty and fear of poverty can seriously damage self-esteem and diminish perceived control over one's life [21] while also affecting confidence in one's ability to cope with adversity [46], which can result in the emergence of depressive symptoms [47].

This result can be interpreted in light of the vulnerability and economic uncertainty experienced by many Chileans [48]. According to OECD estimates [49], more than half of all Chileans are at risk of falling into poverty if they do not receive their salary for three months. Among OECD countries, Chile is where a person from the fourth quintile is most likely to fall back to the first quintile within four years [49]. Likewise, some studies suggest that, in a large percentage of middle-class families, there is an intense "fear of falling": a permanent anxiety over regressing and losing the social status gained in recent decades [48]. This has also been described as an experience of "positional inconsistency" [50], that is, the perception that places and social trajectories are unstable and highly permeable to precarization. In Chile, some studies have shown that this sense of insecurity and uncertainty is associated with depressive symptoms [51].

As is well known, being a woman increases the likelihood of suffering anxiety and depressive symptoms [52]. This gender gap can be related to the fact that women are more often exposed to social disadvantages (lower education and income levels, lower skilled occupations,

power and status inequality) and lifelong stressors [53]. A recent study in Chile reveals that, in confinement conditions, women spend the most time on domestic and childcare tasks, facing an overload of work [54]. Since women are more likely than men to have informal contracts, and thus lack the social protection of formal employment, they can be expected to be more affected by job losses in times of economic instability [9, 10].

Another finding of this study is the fact that living in central Chile increases the risk of psychological distress. This is an expected result, since infection and death numbers have been particularly high in the Metropolitan Region, which includes the capital, Santiago [28, 55]. This is also the region where the lockdown has lasted the longest and where the most restrictive measures to people's mobility have been implemented. This result is consistent with other studies showing that people living in the areas with the strictest confinement measures and the highest rates of COVID-19 infection and mortality report poorer mental health indicators [25, 56].

Finally, this study shows that people with a history of diagnosed mental disorders are more likely to suffer psychological distress than participants without a prior diagnosis. People with pre-existing mental health problems tend to be more reactive to stress [57] and may have less access to services and support since the onset of the COVID-19 outbreak [3, 4], which may contribute to the development or exacerbation of psychological distress. However, this result should be considered with caution, as it is not possible to conclude that the current pandemic and lockdowns are more stressful for those with or at increased risk of mental disorders [5, 13, 58]. For example, a British study showed that adults with pre-existing mental illness diagnoses experienced higher levels of anxious and depressive symptoms during the first weeks of quarantine in the UK; nevertheless, the authors found little difference between the overall mental health trajectories of people with and without diagnosed mental illness [59]. Given their current relevance, these findings need to be further explored through longitudinal strategies.

It is interesting to note that there is a negative association between age and depressive symptoms. This result is consistent with the emergent COVID-19 literature showing higher levels of psychological distress, especially depressive symptoms, among people under 40 years of age [4, 11]. Previous studies have suggested that younger people may be particularly affected by social contact deprivation compared to other age groups, since they are in a period of life characterized by a greater need for interaction with peers [12, 60, 61]. In addition, a proportion of individuals under 35 years of age are college students, who may experience higher levels of depressive symptoms due to the closure of higher education institutions and difficulties in distance learning. Alternatively, it could also be argued that it is more common for older people to find themselves in a situation of reduced mobility or isolation, which may favor their adaptation to lockdown conditions.

Our results disaggregated by sex show that, for men and women, household income was significantly associated with depressive symptoms; however, in the case of women, higher household income was related to higher burden of depressive symptoms, while the reverse association was found for men. One possible interpretation of this result, which should be further analyzed in future studies, is that women might be more concerned about losing the social status associated with their income level.

## Policy implications

The COVID-19 crisis has revealed a series of deficiencies in Chile's health system and public policies. Since some projections suggest that the pandemic may have multiple waves of infection and that periods of lockdown may need to be reintroduced [62], we must rethink the traditional social and healthcare policy response and implement local evidence-based programs [13].

Greater efforts are needed to protect the psychological well-being of people during the pandemic. This requires both mental health and social policies aimed at ensuring access to psychosocial support and reducing people's economic uncertainty.

In Chile, many people have limited access to mental health services [33], which has been made worse by confinement measures. In this scenario, digital technology is an important tool for providing preventive and treatment interventions in mental health [63]. In June 2020, the Chilean government began to implement a program called "Saludable-mente" [Healthy-mind] (https://www.gob.cl/saludablemente/), which is an online platform aimed at delivering psychosocial support to people with mental health problems. These types of programs are relevant, since long-term recovery from the COVID-19 crisis must take into account the potential impact of psychological distress on employment opportunities and economic autonomy.

Social and financial protection measures, such as cash transfer programs or active labor market programs that help people to retain or regain jobs, are needed, particularly within the most vulnerable groups, to reduce socioeconomic insecurity and mitigate the short- and long-term impacts of the crisis on psychological well-being [16, 22, 27, 46]. In this context, cash transfer programs have been implemented in several countries to support individuals and families during the pandemic [27, 64]. Our findings suggest that a gender perspective will be important in these efforts, as women are likely to be the hardest hit by the COVID-19 pandemic [9] and are in a disadvantaged position in our society [29].

## Strengths and limitations

One of the main strengths of this study is that it was based on a nationwide representative sample. Unlike most of the available research on mental health and the COVID-19 pandemic [34], this study has a wide heterogeneity and good stratification in multiple sociodemographic groups. In addition, the analyses were weighted considering basic demographic estimates of the population.

However, an important limitation of this study concerns the data collection strategy used, since it is a self-report instrument that could be affected by the intrinsic limitations of telephone surveys. Nevertheless, this seems to be a better strategy than an online survey, in which participants without Internet access in their homes are not represented. In addition, while we used a well-established and sensitive scale to measure anxious and depressive symptoms (PHQ-4) [36, 40], the results of this study should be considered with caution since it is an ultra-brief self-report instrument.

Another important limitation is related to the cross-sectional nature of the study, which prevents the observation of changes in participants' perceptions and responses to the impact of COVID-19 over time. The lack of a pre-pandemic baseline prevents a direct association between current levels of psychological distress and the specific pandemic context. However, we hope to mitigate some of these problems through the collection of longitudinal data, which will make it possible to explore the long-term predictors of psychological distress stemming from the COVID-19 pandemic.

## Conclusion

The results of this study suggest that being a woman, feeling lonely and isolated, living in the areas hit hardest by the pandemic and lockdown, expecting a lack of income due to having to stop working as a consequence of the pandemic, and having a history of diagnosed mental disorders are significantly associated with psychological distress.

These findings highlight the need to implement social programs to address economic uncertainty and reduce the risk of psychological distress due to the direct and indirect effects

of the COVID-19 pandemic. This study could contribute to identifying what types of psychosocial and economic measures are needed to protect the psychological well-being of the population both during the current pandemic and in preparation for future epidemic outbreaks.

## Supporting information

**S1 Data.**
(DO)

**S2 Data.**
(DTA)

## Acknowledgments

We would like to thank the Centro de Microdatos of the University of Chile.

## Author Contributions

**Conceptualization:** Fabián Duarte, Álvaro Jiménez-Molina.

**Data curation:** Fabián Duarte.

**Formal analysis:** Fabián Duarte.

**Funding acquisition:** Fabián Duarte.

**Investigation:** Fabián Duarte, Álvaro Jiménez-Molina.

**Methodology:** Fabián Duarte, Álvaro Jiménez-Molina.

**Writing – original draft:** Fabián Duarte, Álvaro Jiménez-Molina.

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
