## [Decision Letter · Decision Letter 0]

3 Mar 2021

PONE-D-20-29130

Psychological distress during the COVID-19 epidemic in Chile: the role of economic uncertainty

PLOS ONE

Dear Dr. Jiménez-Molina,

Thank you for submitting your manuscript to PLOS ONE. After careful consideration, we feel that it has merit but does not fully meet PLOS ONE’s publication criteria as it currently stands. Therefore, we invite you to submit a revised version of the manuscript that addresses the points raised during the review process.

Please address the concerns and recommendations by reviewers 1 and 2. Please pay close attention to the following issues: (1) Reviewer 1 states that the main weakness of the study is its cross-sectional nature, and therefore lack of pre-pandemic measures. This weakness could be tackled, somewhat, if the authors quoted in the discussion some comparable statistics, from the same measure, taken before the pandemic. This would ideally come from a Chilean sample, but could come from a demographically similar sample. If this is not available, then this should be said in the discussion. (2) Reviewer 2 states that the adequacy of the sample is poorly described.  What is the statistical rationale for the sample size in terms of power considerations? Please follow up on the issues raised by reviewer 2 with respect to sampling.

We look forward to receiving your revised manuscript.

Kind regards,

M. Harvey Brenner, PhD

Academic Editor

PLOS ONE

2. In the ethics statement in the Methods and online submission information, please clarify whether consent was written or verbal.  If verbal, please also specify: 1) whether the ethics committee approved the verbal consent procedure, 2) why written consent could not be obtained, and 3) how verbal consent was recorded. If the need for consent or parental consent was waived by the ethics committee, please include this information.

Reviewers' comments:

Reviewer's Responses to Questions

**Comments to the Author**

1. Is the manuscript technically sound, and do the data support the conclusions?

Reviewer #1: Yes

Reviewer #2: Partly

2. Has the statistical analysis been performed appropriately and rigorously? 

Reviewer #1: Yes

Reviewer #2: No

3. Have the authors made all data underlying the findings in their manuscript fully available?

Reviewer #1: No

Reviewer #2: Yes

4. Is the manuscript presented in an intelligible fashion and written in standard English?

Reviewer #1: Yes

Reviewer #2: Yes

5. Review Comments to the Author

Reviewer #1: The study presents mental health measured during the Cv19 pandemic using a telephone survey. It is a well written and nicely presented study, and I enjoyed reading it. A key strength of is its setting (Latin America), lending weight to analyses from western European and north American survey’s. Also, the use of probability sampling sets it apart from many surveys undertaken since the pandemic. The main weakness is its cross-sectional nature, and therefore lack of pre-pandemic measures. This weakness could be tackled, somewhat, if the authors quoted in the discussion some comparable statistics, from the same measure, taken before the pandemic. This would ideally come from a Chilean sample, but could come from a demographically similar sample. If this is not available, then this should be said in the discussion.

Minor comments:

The summary of the background literature is clear, although I think you could mention the increased impact that the pandemic has had on young people (see Pierce et al “Mental health before and during the COVID19 pandemic…” Lancet Psychiatry).

Please report all percentages to the same number of decimal places (I suggest 1 dp).

Table 1 would benefit from having the weighted population for comparison

Generally I do not like seeing standard errors next to estimates in tables, they are difficult to interpret confidence intervals are much more preferable.

It is not clear from looking at the table3 or the results section whether the regression models are adjusted.

The finding for household income is interesting but a little hard to interpret: are you saying the richer you get for men the worse the mental health is, and vice versa for women?

It would be good to see age in table 3 broken down in to categories. Again, it’s a little hard to interpret the coefficient just looking at table.

Reviewer #2: The statistical analysis is routine. However, the adequacy of the sample is poorly described. What is the statistical rationale for the sample size in terms of power considerations?

The authors state that the sampling is representative, but abandon any further explanation of this statement. Why is the sample representative of the population? Are proportions of certain representative groups adequate? No detail is described on how the sample was chosen and what factors were adequately provided beforehand for inclusion in the sampling plan. How were strata set? Age groups and gender are considered. What other relevant factors were considered, if any? How exactly were household and education stratified in the sampling?

6. PLOS authors have the option to publish the peer review history of their article (what does this mean?). If published, this will include your full peer review and any attached files.

Reviewer #1: **Yes: **Matthias Pierce

Reviewer #2: No

---

## [Author Response · Author response to Decision Letter 0]

7 Apr 2021

Dear editor and reviewers,

We are grateful for the reviewers’ valuable comments and suggestions, which have allowed us to improve the manuscript. 

In the new version of the manuscript, we include some new references in the background and discussion section. We clarify the verbal consent procedure for participants, and we made some minor corrections to the English. 

We also attach a new file with the new tables. We revised all tables. The income variable in Table 1 was corrected with a new exchange rate between Chilean peso and US dollar. We used the average exchange rate between May and June 2020.

In Table 2, we detected an error in the chi2 of the variable Psychological distress: it said 6.89, but it is 16.78. We also detected an error in the results for expectation increased debt. These changes did not imply changes in the final results of the article.

All changes have been underlined in red in the manuscript (track changes) and some comments have been added when it was necessary to clarify a change made.

We have made the database of the study open access, and we also attach the do file that allows replication of the results.

Here we respond to each of the comments and questions raised by the reviewers (these answers can also be found in the response letter to the reviewers):

1. For comparison with pre-pandemic data, prevalence of depressive symptoms (PHQ-2) was calculated from a previous longitudinal study (ELSOC), which covers the 2016-2018 period and uses a similar community sample in Chile. Unfortunately, data from this survey for 2019 are not yet available.

2. The prevalence of depressive symptoms according to the ELSOC (2016-2018 average) was compared with that of our 2020 study. Prevalence was also compared by sex and age groups.

The average prevalence of depressive symptoms between 2016-2018 (ELSOC) was 20.6%. The prevalence of depressive symptoms in our study in 2020 was 23.3%.

For the 2016-2018 ELSOC, the average prevalence of depressive symptoms by group were:

Women: 24.1% vs Men: 16.9%

Age groups: 18-35: 21.2% ; 36-59: 20.1% ; 60+: 20.6%

In our study (2020), we obtained the following results:

Women: 30.8% vs Men: 15.5%

Age groups: 18-35: 31.5% ; 36-59: 18.5% ; 60+: 18.5%

These results suggest that there may have been an increased burden of depressive symptoms, especially among women and young people. These results were integrated into the discussion, pointing out the limitations of the comparison as they are different samples.

3. The impact of the pandemic on young people's mental health was included in the background section, using the reference noted by the reviewer. This point is discussed in more detail in the discussion.

4. Weighted population data were included in table 1.

5. Expansion factors were included in the calculations of all the regression models and tables, except table 2.

6. We ran a regression using dummy variables for age categories. The results are similar in general. Regarding age, we see an increase in anxiety for age group 60+, a decrease in depression for age group 36-59 and 60+, and no effect on psychological distress. The omitted variable is age group 18-35 (See Table 3)

7. We indicate all percentages to the same number of decimal places, as suggested.

8. We include confidence intervals in each table.

9. Reviewer #1. The finding for household income is interesting but a little hard to interpret: are you saying the richer you get for men the worse the mental health is, and vice versa for women? 

Authors: No, the interpretation is the opposite: for men, higher household income is related to lower burden of depressive symptoms. For women, higher household income is related to higher burden of depressive symptoms. We have rewritten the interpretation of this result in the discussion section to make it clearer.

10. The sample size was selected to get absolute sampling errors below 5% for a proportion of 0.5 and assuming a simple random sampling, when dividing the sample into men and women.

The sampling frame arises from a probabilistic, geographically stratified, and multistage sampling, where "municipalities" are selected in the first stage, "census blocks" in the second stage, "occupied dwellings" in the third stage, and "persons aged 18 years and over”, in the fourth stage according to the random method. The first part of the Method describes the multi-stage sampling process.

Representativeness is achieved for two reasons:

1 First, the probability of selection of each surveyed unit is known, since the sample design is probabilistic (two-phase), where in a first stage a survey is carried out to collect telephone numbers, based on a census sampling frame. From the surveys obtained in this stage, a second sampling is carried out where telephone calls are made to carry out the final survey.

2 Second, the sample size reached ensures sampling errors within the standards. Regarding the sample errors, they were computed considering a simple random sampling, for a proportion of 0.5. The following levels are reached:

1.a National, with an absolute error of 2.98%.

1.b Men and women with absolute errors of 4.31% and 4.14%, respectively.

1.c Age Group 15-35, 36-59 and 60+ years old, with relative errors between 5.77%, 4.49% and 5.57%, respectively.

Strata were not defined in the sample design, but post-stratification adjustments were included in the expansion factors to reach the population values in the groups of men and women, and for the age groups of 15-35, 36-59, and 60+ years.

Another element that was considered in the calculations of the expansion factors is the inclusion of telephone non-response rates, to include possible effects of the survey method.

11. The sampling design did not include household or education stratification. Post-stratification adjustments were included to estimate the expansion factors.

---

## [Decision Letter · Decision Letter 1]

3 May 2021

Psychological distress during the COVID-19 epidemic in Chile: the role of economic uncertainty

PONE-D-20-29130R1

Dear Dr. Jiménez-Molina,

We’re pleased to inform you that your manuscript has been judged scientifically suitable for publication and will be formally accepted for publication once it meets all outstanding technical requirements.

Kind regards,

M. Harvey Brenner, PhD

Academic Editor

PLOS ONE

Additional Editor Comments (optional):

Reviewers' comments:

Reviewer's Responses to Questions

**Comments to the Author**

1. If the authors have adequately addressed your comments raised in a previous round of review and you feel that this manuscript is now acceptable for publication, you may indicate that here to bypass the “Comments to the Author” section, enter your conflict of interest statement in the “Confidential to Editor” section, and submit your "Accept" recommendation.

Reviewer #2: All comments have been addressed

2. Is the manuscript technically sound, and do the data support the conclusions?

Reviewer #2: (No Response)

3. Has the statistical analysis been performed appropriately and rigorously? 

Reviewer #2: (No Response)

4. Have the authors made all data underlying the findings in their manuscript fully available?

Reviewer #2: (No Response)

5. Is the manuscript presented in an intelligible fashion and written in standard English?

Reviewer #2: (No Response)

6. Review Comments to the Author

Reviewer #2: (No Response)

7. PLOS authors have the option to publish the peer review history of their article (what does this mean?). If published, this will include your full peer review and any attached files.

Reviewer #2: No

---

## [Editor Report · Acceptance letter]

21 Oct 2021

PONE-D-20-29130R1 

Psychological distress during the COVID-19 epidemic in Chile: the role of economic uncertainty 

Dear Dr. Jiménez-Molina:

I'm pleased to inform you that your manuscript has been deemed suitable for publication in PLOS ONE. Congratulations! Your manuscript is now with our production department. 

Kind regards, 

on behalf of

Professor M. Harvey Brenner 

Academic Editor

PLOS ONE